# DDFNet-A: Attention-Based Dual-Branch Feature Decomposition Fusion Network for Infrared and Visible Image Fusion

**Qiancheng Wei [1], Ying Liu [1,\*], Xiaoping Jiang [1], Ben Zhang [1], Qiya Su [2] and Muyao Yu [2]**

[1] University of Chinese Academy of Sciences, Beijing 101408, China; weiqiancheng17@mails.ucas.ac.cn (Q.W.); jiangxiaoping17@mails.ucas.ac.cn (X.J.); zhangben23@mails.ucas.ac.cn (B.Z.)

[2] Beijing Institute of Remote Sensing Equipment, Beijing 100005, China; qysjes@163.com (Q.S.); yumuyao25s@163.com (M.Y.)

\* Correspondence: yingliu@ucas.ac.cn

**Abstract:** The fusion of infrared and visible images aims to leverage the strengths of both modalities, thereby generating fused images with enhanced visible perception and discrimination capabilities. However, current image fusion methods frequently treat common features between modalities (modality-commonality) and unique features from each modality (modality-distinctiveness) equally during processing, neglecting their distinct characteristics. Therefore, we propose a DDFNet-A for infrared and visible image fusion. DDFNet-A addresses this limitation by decomposing infrared and visible input images into low-frequency features depicting modality-commonality and high-frequency features representing modality-distinctiveness. The extracted low and high features were then fused using distinct methods. In particular, we propose a hybrid attention block (HAB) to improve high-frequency feature extraction ability and a base feature fusion (BFF) module to enhance low-frequency feature fusion ability. Experiments were conducted on public infrared and visible image fusion datasets MSRS, TNO, and VIFB to validate the performance of the proposed network. DDFNet-A achieved competitive results on three datasets, with EN, MI, VIFF, $Q^{AB/F}$, FMI, and $Q_s$ metrics reaching the best performance on the TNO dataset, achieving 7.1217, 2.1620, 0.7739, 0.5426, 0.8129, and 0.9079, respectively. These values are 2.06%, 11.95%, 21.04%, 21.52%, 1.04%, and 0.09% higher than those of the second-best methods, respectively. The experimental results confirm that our DDFNet-A achieves better fusion performance than state-of-the-art (SOTA) methods.

**Keywords:** infrared image; visible image; image fusion; multi-modality; attention

## 1. Introduction

Infrared and visible imaging sensors offer distinctive modalities for targets, and their fusion is a crucial research topic in the fields of computer vision and image processing [1,2]. Infrared images highlight the thermal radiation emitted by objects through pixel intensity. However, they often have lower resolution and lack detailed texture information. However, visible images present rich textual detailed features through gradients and edges. However, they face challenges in providing valuable information about objects under weak light conditions. The fusion of complementary features from infrared and visible modalities into a fused image can provide a more accurate scene depiction than any single modality image, thus further supporting advanced visual tasks. The fusion of infrared and visible images plays a crucial role in various fields such as remote sensing [3,4], object tracking [5,6], object detection [7], and other fields requiring comprehensive scene understanding. He et al. proposed a multi-level fusion algorithm [8] that enhances target visibility by integrating both pixel- and feature-level fusion, effectively addressing the relationship between low- and high-frequency components. Similarly, Stephen et al. introduced an improved target tracking method [9] through infrared–visible image fusion, employing a PCA-weighted fusing rule. Surveillance systems capitalize on the complementary properties of infrared

and visible images, employing fusion techniques such as framelet transform [10], NSCT decomposition [11], and hybrid fusion algorithms [12] for enhanced object detection. In the domain of remote sensing, Li et al. proposed a medium-altitude unmanned aerial vehicle remote sensing system [13] integrating image registration and fusion. Additionally, fusion techniques have been applied to geostationary meteorological satellite image fusion [14].

The challenge in designing an effective fusion algorithm lies in its ability to extract and retain useful information from both modalities, while simultaneously eliminating redundancy and noise. This process requires a delicate balancing act to enhance the valuable attributes of the images without introducing unnecessary information or distortion. According to the variances in fusion theory and strategy, infrared and visible image fusion (IVIF) techniques can be divided into two types: traditional and deep learning-based methods. Traditional IVIF methods follow a three-stage pipeline: feature extraction employing hand-crafted models, feature fusion using specific strategies, and image reconstruction using an inverse feature extractor. These methods are categorized into multi-scale transformation-based methods, sparse representation-based methods, saliency detection-based methods, and spatial transformation-based methods. Multi-scale transform-based fusion methods can improve visual attention expression, thereby achieving better fusion results [15,16]. Sparse representation is applied in image fusion [17,18] because it can fully preserve the information of the source images in the fusion result by using the learning ability of the over-complete dictionary. Saliency-based fusion methods extract important features (saliency maps) from source images and integrate them with the transformed source data to generate fused images. Spatial transformation-based fusion methods project high-dimensional source images into a lower-dimensional space for feature analysis, facilitating effective fusion [19]. Although traditional methods have achieved good results in the IVIF task, they also exhibit some limitations. First, traditional methods often use identical transformations for feature extraction from diverse source images, neglecting their unique characteristics and limiting the feature representation. Second, manually designed fusion strategies in traditional methods often face challenges when adapting to the increasing complexity of fusion tasks, thereby limiting their overall performance.

Recently, advancements in deep learning have partially addressed the limitations of traditional methods in IVIF tasks [20]. First, deep learning-based methods can harness dual-branch network architectures to achieve distinct feature extraction for each source image, thereby capturing more focused and informative features. Second, deep learning excels in image fusion by leveraging networks for adaptive feature integration and continually refining the results through suitable loss functions. Deep learning-based frameworks for infrared and visible image fusion can be broadly categorized into three types: autoencoders (AEs), convolutional neural networks (CNNs), and generative adversarial networks (GANs). In addition, transformer-based architectures have emerged as a promising approach for image fusion tasks, gaining increasing attention in recent times. AE-based frameworks leverage an encoder–decoder architecture. The encoder extracts an informative feature representation of the input image pairs, and the decoder then utilizes these extracted features to reconstruct the fused images [21]. CNN-based fusion methods introduce convolutional neural networks into image fusion tasks, leveraging end-to-end feature extraction, fusion, and image reconstruction [22]. GAN-based fusion methods use a competitive training process in which a generator creates fused images, and a discriminator tries to identify real source images from the generated images [23].

However, current IVIF methods have two limitations. First, they often treat common features between modalities (modality-commonality) and unique features from each modality (modality-distinctiveness) equally during feature extraction and fusion, neglecting their distinct characteristics. Second, these methods are limited to effectively extracting modality-distinctiveness features and fusing modality-commonality features. These shortcomings hinder the performance of IVIF tasks [24]. From a frequency domain perspective, to consider the modality characteristics, modality-commonality features are reflected by global low-frequency information, while modality-distinctiveness features are reflected

by local high-frequency information. These features, which have different characteristics, require different fusion strategies. Low-frequency feature fusion prioritizes the interaction between features from infrared and visible images. In contrast, high-frequency feature fusion enhances the representation of local detail information.

Therefore, we propose an attention-based dual-branch feature decomposition fusion network (DDFNet-A) for IVIF tasks. This novel framework decomposes infrared and visible images into low-frequency features, capturing the commonalities across modalities, and high-frequency features, representing the distinctiveness of each modality. These features are then fused and used to reconstruct the final fused images. Our contributions can be summarized in three aspects:

- We propose a novel attention-based dual-branch framework for IVIF. This framework considers the characteristics of each modality during the feature processing phase, enhancing the extraction and fusion capabilities specific to these modality characteristic features.
- We propose a hybrid attention block (HAB) to extract high-frequency features. This block can dynamically adjust the weights of the feature maps across the channel, frequency, and spatial dimensions based on their significance to the task.
- We propose a base feature fusion (BFF) module to fuse low-frequency features. This module utilizes three-stage fusion strategies to integrate the features of the cross-modality global dependencies.
- Experiments on the MSRS [25], TNO [26], and VIFB [27] datasets demonstrated that DDFNet-A achieved superior results in both visual quality and quantitative evaluation.

## 2. Related Work

### 2.1. Deep Learning Based IVIF Methods

AE-based IVIF methods typically employ an encoder–decoder architecture. The encoder extracts informative features from infrared and visible source images, whereas the decoder reconstructs the fused images. Xu et al. [28] used two encoder–decoder networks for feature extraction and decomposition. The extracted features are then fused using weighted averaging and max-pooling strategies. Wang et al. [29] addressed the limitations of missing multi-scale features and weak capture of global dependencies by incorporating dense connections within their network architecture. Zhao et al. [30] decomposed an image into background and detailed features within the encoder. L1-norm attention is employed for weight allocation during this decomposition. The decoder then reconstructs the final image. Li et al. [31] proposed a cross attention mechanism (CAM) for image fusion, employing a two-stage training strategy with auto-encoders for each modality and a CAM decoder to integrate features and enhance fused images. Ji et al. [32] proposed MRANet, a fusion network combining convolutional residual structures with an attention-based multi-scale fusion strategy to effectively extract local and global features from images. Luo et al. [33] proposed a hierarchical fusion network with triple fusion and a cascading edge-prior branch for infrared and visible image fusion, along with a novel loss function for improved edge representation. Wang et al. [34] proposed a two-stream auto-encoder for image fusion, using wavelet decomposition and structural feature map decomposition (SFMD) for enhanced feature fusion with carefully crafted rules.

CNN-based IVIF methods involve end-to-end network modeling for feature extraction, fusion, and image reconstruction. Hou et al. [35] improved the loss function to preserve the local features from the source images to the fused images based on saliency features. STDFusionNet [36] uses saliency information from the source images in the loss function to guide the fusion process. This approach ensures that the fused images retain important information from the infrared images. Tang et al. [25,37] proposed a real-time image fusion network called semantic-aware fusion (SeAFusion) and a progressive image fusion network based on illumination perception (PIAFusion). Li et al. [38] introduced an unsupervised fusion model for infrared and visible images. It employs residual dense blocks for feature extraction and gradient loss to enhance texture detail expression. Pan et al. [39] introduced

an efficient VIF model based on CNN, preserving details and integrating feature information adaptively guided by source images within the learning model. Yang et al. [40] introduced a fusion framework merging multi-scale CNNs with saliency weight maps. Initial weights from source features are refined, processed by CNNs, and adjusted adaptively using saliency. Tang et al. [41] proposed EdgeFusion, a method combining weighted least squares decomposition, sub-window variance filtering, and visual saliency mapping for infrared and visible image fusion.

The GAN framework consists of a generator and a discriminator. FusionGAN [42] proposed an adversarial training scheme for the IVIF problem; however, it exhibited limitations in simultaneously preserving crucial information from infrared images and detailed textures from visible images. Ma et al. [43] proposed improvements to address the shortcomings in preserving image details during fusion by incorporating new loss functions into their network. Ma et al. [23] proposed an end-to-end fusion model using a conditional GAN with a dual-discriminator to fuse source images of varying resolutions. Rao et al. [44] proposed AT-GAN, a method for multi-modal image feature extraction using intensity attention, semantic transition, and quality assessment modules to preserve key features and filter noise. Huang et al. [45] proposed MAGAN for fusing infrared and visible images, employing a multiattention generator and two discriminators to preserve salient targets and texture information. Li et al. [46] proposed DANT-GAN, which uses dual attention mechanisms for feature extraction and fusion at local and global levels to preserve information and compensate for feature extraction loss.

### 2.2. Vision Attention Methods

Vision attention mechanisms selectively emphasize important information while suppressing irrelevant details. Methods include channel attention mechanisms, spatial attention mechanisms, and transformer-based approaches.

The channel attention mechanism enhances focus on important channels and reduces focus on unimportant channels by calculating the weight of each channel. SENet [47] utilizes squeeze-and-excitation (SE) blocks to capture global image information and improve the feature representation for object classification tasks. To improve its ability to capture complex features, Gao et al. [48] proposed an enhancement of the SENet architecture by incorporating global second-order pooling (GSoP) within the squeeze module.

The spatial attention mechanism highlights important regions in the feature maps while suppressing the background noise. Mnih et al. [49] proposed a recurrent attention model (RAM), that combined recurrent neural networks (RNNs) with reinforcement learning. This approach allows RAM to focus on informative regions within CNN feature maps, thereby optimizing computational efficiency. Jaderberg et al. [50] proposed a spatial transformer network (STN), that utilizes a sub-network to predict affine transformations, allowing it to focus on important regions within an image.

Furthermore, some methods combine both channel and spatial attention mechanisms to achieve more effective attention. Woo et al. [51] proposed the CBAM module, which sequentially calculates the channel-wise and spatial attention. These attentions are then combined to generate a final attention map, which refines the original feature map by applying weighted adjustments.

The transformer model utilizes self-attention mechanisms to capture global feature dependencies, demonstrating effectiveness particularly in natural language processing (NLP) tasks [52]. Recently, the transformer model has demonstrated potential in computer vision [53,54]. Wu et al. [55] designed a lite transformer (LT) model specifically for mobile devices used in NLP tasks. Their approach reduces the computational load without sacrificing the performance by employing a special group of attention mechanisms that focus on both the local context and long-range dependencies within the text. Restormer [56] combined multi-Dconv head transposed attention (MDTA) and gated-Dconv feed-forward network (GDFN) modules to efficiently capture both local and non-local pixel interactions.

This allows high-resolution images to be handled and achieves excellent performance in image restoration tasks.

Building on the aforementioned research, we propose DDFNet-A, which integrates a hybrid attention mechanism, combining channel, frequency, and spatial attention to extract high-frequency local features. Additionally, our model adopts the LT and Restormer modules to capture global features and generate high-quality fused images.

## 3. Methodology

### 3.1. Overview

The DDFNet-A consists of three components: (1) an encoder for feature extraction, (2) a fusion network for integrating cross-modality features, and (3) a decoder for generating the fused images. The overall framework of the DDFNet-A is illustrated in Figure 1. For simplicity, low-frequency features are referred to as base features, and high-frequency features are referred to as detail features in the subsequent discussion.

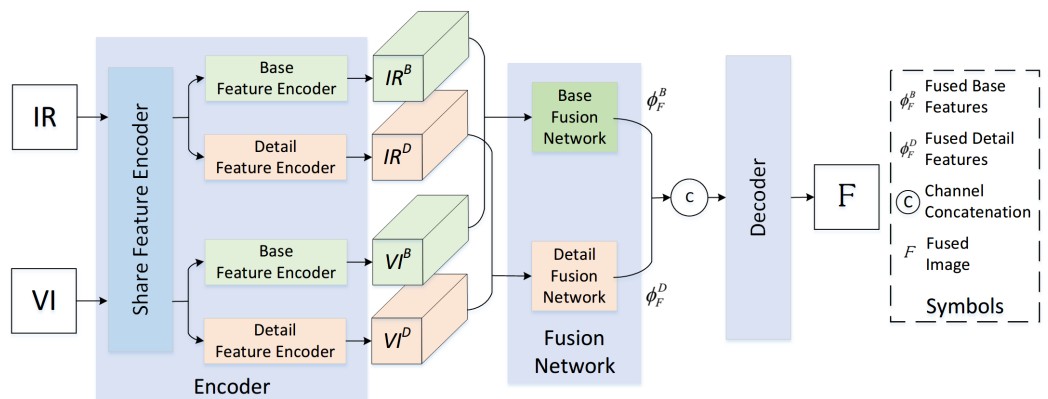

**Figure 1.** The architecture of the DDFNet-A.

### 3.2. Encoder

The encoder in Figure 1 comprises three components: a share feature encoder (SFE), base feature encoder (BFE), and detail feature encoder (DFE). SFE, BFE, and DFE are denoted as $\mathcal{S}(\cdot)$, $\mathcal{B}(\cdot)$, and $\mathcal{D}(\cdot)$, respectively.

#### 3.2.1. Share Feature Encoder

SFE is employed to extract complementary and shared shallow features $\{\phi_{IR}^S, \phi_{VI}^S\}$ from the input infrared and visible images $\{IR, VI\}$, i.e.,

$$\phi_{IR}^S = \mathcal{S}(IR), \ \ \phi_{VI}^S = \mathcal{S}(VI). \tag{1}$$

The Restormer block [56] was selected as the foundational unit of the SFE to derive global context features from input images through self-attention across feature dimensions. This facilitates the extraction of shallow cross-modality features without imposing a significant increase in the computational burden.

#### 3.2.2. Base Feature Encoder

To extract the base features $\{\phi_{IR}^B, \phi_{VI}^B\}$ from shallow features $\{\phi_{IR}^S, \phi_{VI}^S\}$, a lite transformer (LT) block [55] is selected as the foundational unit of the BFE, i.e.,

$$\phi_{IR}^B = \mathcal{B}(\phi_{IR}^S), \ \ \phi_{IR}^B = \mathcal{B}(\phi_{VI}^S). \tag{2}$$

The LT block efficiently manages long-range spatial dependencies and captures overall structural and global background information. Additionally, it reduces the size of the embedding layer while maintaining the performance and computational efficiency by replacing the feedforward neural network with a broader attention layer.

### 3.2.3. Detail Feature Encoder

In contrast to the BTE, the DFE is employed to extract detail features $\{\phi_{IR}^D, \phi_{VI}^D\}$ from shallow features $\{\phi_{IR}^S, \phi_{VI}^S\}$, i.e.,

$$\phi_{IR}^D = \mathcal{D}(\phi_{IR}^S), \ \phi_{VI}^D = \mathcal{D}(\phi_{VI}^S). \tag{3}$$

Similar to RevNets [57], the DFE divides the detail features $\{\phi_{IR}^D, \phi_{VI}^D\}$ channels into two parts, treating them separately and then combining them through a channel concatenation operation. The DFE architecture is shown in Figure 2. This strategy substantially reduces the computational load of the model, while minimally affecting its performance. Moreover, it enhances the ability to capture critical feature information during propagation, thereby improving the detail feature representation for image reconstruction. The calculation process for DFE is as follows:

$$\phi_{I,n+1}^D[c+1:C] = \phi_{I,n}^D[c+1:C] + \mathcal{HAB}_1(\phi_{I,n+1}^D[1:c]),$$
$$\phi_{I,n+1}^D[1:c] = \phi_{I,n}^D[1:c] \odot exp(\mathcal{HAB}_2(\phi_{I,n+1}^D[c+1:C])) + \mathcal{HAB}_3(\phi_{I,n+1}^D[c+1:C]), \tag{4}$$
$$\phi_{I,n+1}^D = \mathcal{CAT}\{\phi_{I,n+1}^D[1:c], \phi_{I,n+1}^D[c+1:C]\}.$$

where $I$ denotes the input image pairs $\{IR, VI\}$, $\phi_{I,n}^D[1:C] \in \mathbb{R}^{h \times w \times C}$ is the features from 1st to the $C$th channels of the input features for the $n$th layer of the DFE. The $\odot$ denotes the Hadamard product, and $\mathcal{CAT}(\cdot)$ denotes a channel concatenation operation. $\mathcal{HAB}_i(\ i=1,2,3)(\cdot)$ is a hybrid attention block (HAB).

The HAB was proposed to serve as the computing unit within the DFE to extract local detail features. The HAB integrates a channel attention block (CAB) with a frequency channel attention block (FCAB) and a spatial attention block (SAB) to enhance detail features and suppress noise. The HAB can learn the importance of features across channels, frequencies, and spatial dimensions. This allows the extraction of more informative feature representations. The structure of the HAB is illustrated in Figure 2.

### 3.3. Fusion Network

The fusion network shown in Figure 1 comprises a base feature fusion (BFF) layer and detail feature fusion (DFF) layer. The BFF and DFF are denoted by $\mathcal{F}^B(\cdot)$ and $\mathcal{F}^D(\cdot)$, respectively.

### 3.3.1. Base Feature Fusion

The BFF is proposed to fuse the base features $\{\phi_{IR}^B, \phi_{VI}^B\}$ to generate fused base features $\phi_F^B$, i.e.,

$$\phi_F^B = \mathcal{F}^B(\phi_{IR}^B, \phi_{VI}^B). \tag{5}$$

The BFF adopts a three-stage fusion strategy to ensure the comprehensive fusion of multi-modality base features. The structure of the BFF is shown in Figure 3. The first stage leverages both addition and concatenation operations to generate the preliminary fused features. Subsequently, an LT block with self-attention was employed in the second stage to capture the global dependencies within the fused features. Finally, the third stage utilizes another LT block with cross-attention to further enrich and enhance the robustness of the fused feature representations. The BFF was calculated as follows:

$$Fusion\,stage\,1: \ \phi_{Add}^B = \phi_{IR}^B + \phi_{VI}^B, \ \phi_{Concat}^B = \mathcal{CAT}(\phi_{IR}^B, \phi_{VI}^B);$$
$$Fusion\,stage\,2: \ \phi_F^{B'} = LT_{Self-Attention}(\phi_{Add}^B + \phi_{Concat}^B); \tag{6}$$
$$Fusion\,stage\,3: \ \phi_F^B = LT_{Cross-Attention}((\phi_F^{B'} \otimes \phi_{Add}^B + \phi_F^{B'} \otimes \phi_{Concat}^B), \ (\phi_{Add}^B + \phi_{Concat}^B)).$$

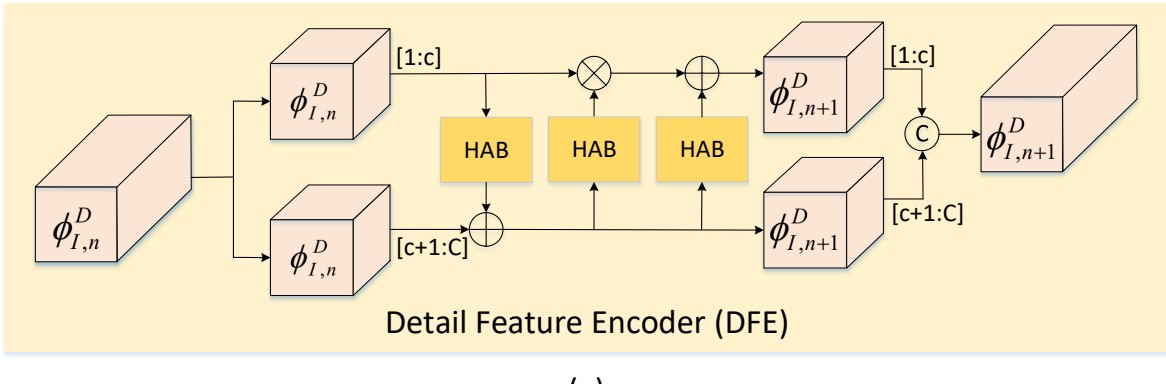

(a)

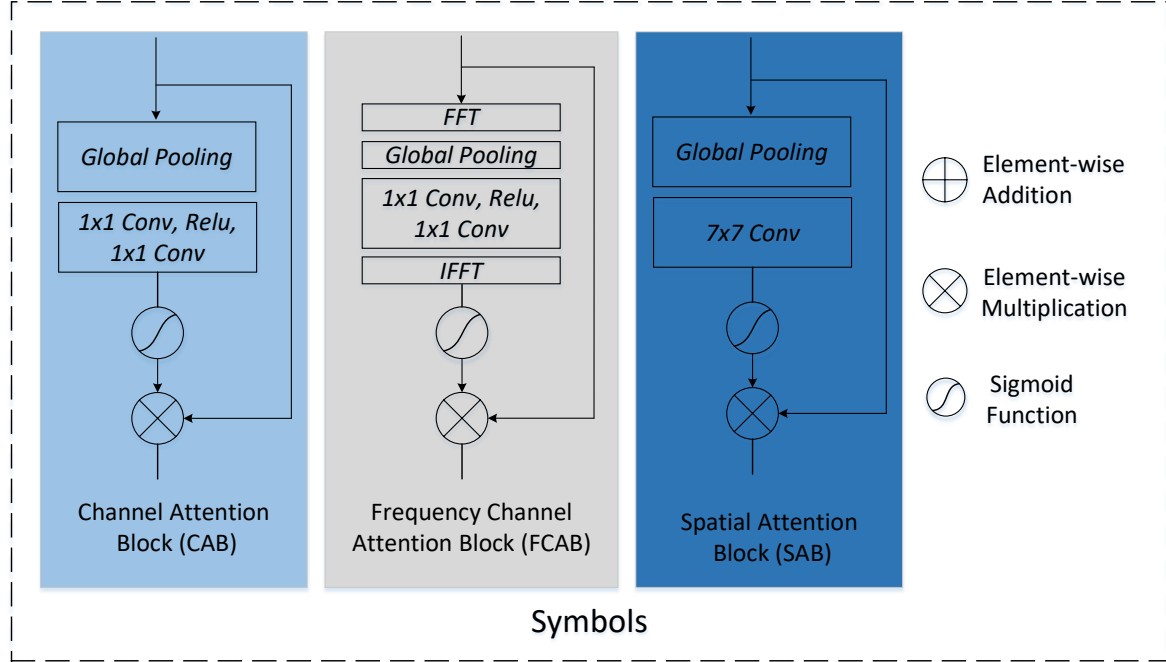

**Figure 2.** The architecture of the DFE and HAB.

In the calculation of cross-attention in the third stage of fusion, $(\phi_F^{B'} \otimes \phi_{Add}^B + \phi_F^{B'} \otimes \phi_{Concat}^B)$ serves as the query, while $(\phi_{Add}^B + \phi_{Concat}^B)$ serves as both the key and the value. A residual structure is employed to prevent information loss.

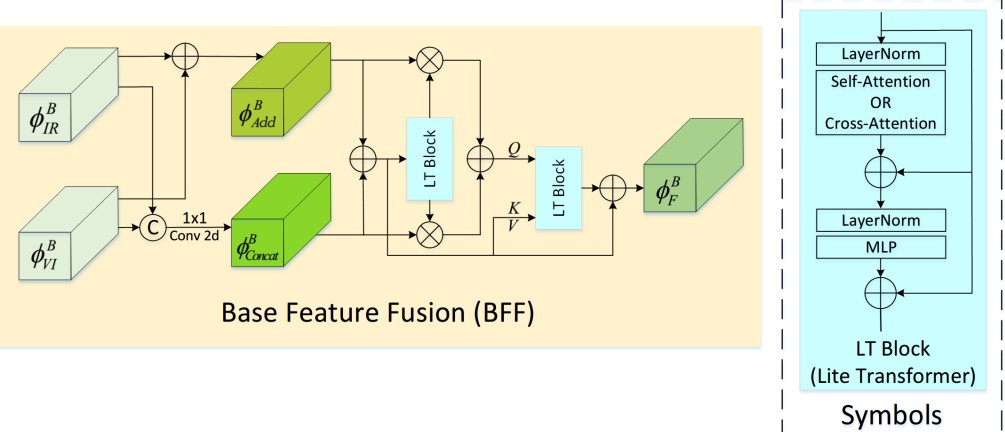

**Figure 3.** The architecture of the BFF.

3.3.2. Detail Feature Fusion

The DFF is employed to fuse the detail features $\{\phi_{IR}^D, \phi_{VI}^D\}$ to generate fused detail features $\phi_F^D$, i.e.,

$$\phi_F^D = \mathcal{F}^D(\phi_{IR}^D, \phi_{VI}^D). \tag{7}$$

Considering that the inductive bias for DFF should be similar to that of DFE, we employed a DFE-like network for the DFF layer.

*3.4. Decoder*

The fused base features and detail features are concatenated along the channel dimension to form the input of decoder $\mathcal{DR}(\cdot)$ in Figure 1, which then outputs the fused images $F$.

$$F = \mathcal{DR}(\mathcal{CAT}(\phi_F^B, \phi_F^D)). \tag{8}$$

Considering the characteristics of the fused base and detail features, similar to SFE, we selected the Restormer module as the foundational unit for the decoder.

*3.5. Loss Function*

In the image fusion task, ground truth is not available for supervision. For image decomposition and reconstruction, a loss function for model training was proposed as follows:

$$Loss = L_1 + \alpha_1 L_2 + \alpha_2 L_3. \tag{9}$$

where $L_1$ is the reconstruction loss, which measures the difference between input images $I$ and fused images $F$. $L_2$ is the decomposition loss used for decomposing the input images into base and detail features. $L_3$ is the gradient loss used to preserve the texture details. Hyperparameters $\alpha_1$ and $\alpha_2$ are used to adjust the weights of $L_2$ and $L_3$. $L_1$ comprises the Charbonnier loss [58] and SSIM loss [59]. Charbonnier loss is used for detail reconstruction, whereas SSIM loss measures the difference in structure between the input and fused images.

$$
\begin{aligned}
L_1 &= L_{Charbonnier} + \beta L_{SSIM} \\
L_1 &= \sqrt{||\max(IR, VI) - F||^2 + \varepsilon_1^2} + \beta(1 - SSIM(I, F)).
\end{aligned} \tag{10}
$$

where $\beta$ is used to adjust the weights of $L_{SSIM}$ and $\varepsilon_1 = 10^{-3}$ is a small constant, following Lai and Huang [58]. CC loss [24] is adopted for feature decomposition, leveraging its ability to decompose the input images based on modality correlation.

$$L_2 = \frac{(L_{cc}^D)^2}{L_{cc}^B} = \frac{(cc(\phi_{IR}^D, \phi_{VI}^D))^2}{cc(\phi_{IR}^B, \phi_{VI}^B) + \varepsilon_2}. \tag{11}$$

where $cc(\cdot)$ denotes the correlation coefficient, and $\varepsilon_2$ is set to 1.01 to ensure that this term remains positive at all times, following Zhao and Bai [24]. Gradient loss $L_3$ aggregates the brightness and edge information of the input images, preserving the texture details.

$$L_3 = \frac{1}{HW} || |\nabla_F| - \max(|\nabla_{IR}|, |\nabla_{VI}|)||_1. \tag{12}$$

where $\nabla$ represents the Sobel gradient operator, $H$ and $W$ are the height and width of the images.

## 4. Experimental Results and Analysis

To verify the performance of the DDFNet-A, we conducted extensive experiments on the public datasets. First, we introduce experimental settings, datasets, and evaluation metrics. Then, we conducted an ablation study and comparative experiments to validate the method proposed in this paper. The experimental details are outlined below.

### 4.1. Experiment Setup

Experiments were conducted on a machine equipped with eight NVIDIA V100 GPUs. During the preprocessing stage, each training sample was randomly cropped into patches of size 128 × 128. Training was performed for 30 epochs, with a batch size of 32. The Adam optimizer was employed with a learning rate of $10^{-4}$. For the loss functions in Equations (9) and (10), the values of $\alpha_1$, $\alpha_2$, and $\beta$ were set to 2, 1, and 5, respectively, determined through a grid search for optimal performance. For the network hyperparameters setting, the number of Restormer blocks in SFE is 4, with 8 attention heads and 64 dimensions. The dimension of the LT block in BTE is also 64 with 8 attention heads. The configuration of the decoder is the same as the encoder.

### 4.2. Dataset

IVIF experiments were conducted using three widely recognized datasets, i.e., MSRS, TNO, and VIFB. DDFNet-A was trained on the MSRS training set consisting of 1083 pairs of infrared and visible images. The MSRS test set with 361 pairs, TNO with 25 pairs, and VIFB with 18 pairs were used as test datasets to evaluate the fusion performance.

The MSRS dataset describes traffic scenes, including various objects such as cars, pedestrians, and bicycles, in diverse environments including daytime and nighttime. It comprises 1444 pairs of high-quality aligned infrared and visible images with a resolution of 480 × 640. Additionally, an image enhancement algorithm based on dark channel prior is utilized to optimize the contrast and signal-to-noise ratio of the infrared images.

The TNO dataset comprises 63 pairs of infrared and visible images captured in diverse military and surveillance scenarios, with varying resolutions preserved. These images encompass enhanced visual, near-infrared, and long-wave infrared spectra, showcasing different objects and targets against backgrounds such as rural and urban areas.

The VIFB dataset is a test set comprising 21 pairs of visible and infrared images with varying resolutions. These image pairs are sourced from the Internet and various tracking datasets, encompassing diverse environments and conditions like indoor, outdoor, low illumination, and over-exposure. As a result, the dataset serves to evaluate the generalization capability of image fusion algorithms.

### 4.3. Evaluation Metrics

To evaluate the image fusion capability of the model, eight metrics were selected: entropy (EN) [60], mutual information (MI) [61], visual information fidelity for fusion (VIFF) [62], gradient-based fusion metric ($Q^{AB/F}$) [63], average gradient (AG) [64], feature mutual information (FMI) [65], salient feature information ($Q_s$) [66], and structural sim-

ilarity index measure (SSIM) [59]. Higher values of these metrics generally correspond to superior image fusion quality. These metrics evaluate various aspects of the fused images, including the information content, preservation of source image information, and visual quality.

EN measures the information content and uncertainty within an image based on the distribution of pixel values. The EN is defined as follows:

$$EN = -\sum_{i=0}^{L-1} p_i \log_2 p_i. \tag{13}$$

where $L$ denote the number of gray levels, and $p_i$ represent the normalized histogram of the corresponding gray level in the fused image. A higher entropy value generally indicates a richer information content, though it is essential to consider any noise introduced during the fusion process.

MI quantifies the correlation between the fused images and original images, indicating how well the fused images capture information from both sources.

$$MI = MI_{A,F} + MI_{B,F}. \tag{14}$$

where $MI_{A,F}$ and $MI_{B,F}$ represent the amount of information transferred from the infrared and visible images to the fused image, respectively. Mutual information ($MI$) between two random variables can be calculated using the Kullback–Leibler measure, defined as follows:

$$MI_{X,F} = \sum_{x,f} P_{X,F}(x,f) \log \frac{P_{X,F}(x,f)}{P_X(x)P_F(f)}. \tag{15}$$

where $P_X(x)$ and $P_F(f)$ represent the marginal histograms of the source image $X$ and the fused image $F$, respectively. $P_{X,F}(x,f)$ denotes the joint histogram of the source image $X$ and the fused image $F$. A high MI metric suggests significant transfer of information from the source images to the fused image, indicating effective fusion performance.

VIFF evaluates the preservation of visual information in the fused images compared with the source images by considering brightness, contrast, structure, and texture. VIFF is defined as follows:

$$VIFF_{X,F}^{VI} = \frac{\sum\limits_{i \in subbands} I(\vec{C}, \vec{F}|R^{S,i})}{\sum\limits_{i \in subbands} I(\vec{C}, \vec{X}|R^{S,i})} \tag{16}$$

where $I(\vec{C}, \vec{F}|R^{S,i})$ and $I(\vec{C}, \vec{X}|R^{S,i})$ represent the ideal information extracted by the human brain from the source image and the fusion image, respectively. A higher VIFF indicates better visual quality.

$Q^{AB/F}$ estimates the performance of salient information from the inputs within the fused images using local gradient measurements. $Q^{AB/F}$ is defined as follows:

$$Q^{AB/F} = \frac{\sum_{i=1}^{N} \sum_{j=1}^{M} Q^{AF}(i,j)\omega^A(i,j) + Q^{BF}(i,j)\omega^B(i,j)}{\sum_{i=1}^{N} \sum_{j=1}^{M} (\omega^A(i,j) + \omega^B(i,j))}. \tag{17}$$

where $Q^{XF}(i,j) = Q_g^{XF}(i,j)Q_a^{XF}(i,j)$, where $Q_g^{XF}(i,j)$ and $Q_a^{XF}(i,j)$ represent the edge strength and orientation values at location $(i,j)$, respectively. $\omega^X$ signifies the weight expressing the importance of each source image to the fused image. A high $Q^{AB/F}$ suggests significant transfer of edge information to the fused image.

AG calculates the average gradient value across all pixels, representing the overall spatial variation (sharpness) in the images.

$$AG = \frac{1}{MN} \sum_{i=1}^{M} \sum_{j=1}^{N} \sqrt{\frac{\nabla F_x^2(i,j) + \nabla F_y^2(i,j)}{2}}. \tag{18}$$

where $\nabla Fx(i,j) = F(i,j) - F(i+1,j)$ and $\nabla Fy(i,j) = F(i,j) - F(i,j+1)$. A higher AG metric suggests a greater presence of gradient information in the fused image, indicating improved performance of the fusion algorithm.

FMI quantifies the shared information between the features extracted from the fused images and the original images, assessing the preservation of relevant details. The AG is defined as

$$FMI = MI_{A',F'} + MI_{B',F'}. \tag{19}$$

where $A'$, $B'$, and $F'$ represent the feature maps of the infrared, visible, and fused images, respectively. A significant FMI metric typically suggests substantial transfer of feature information from the source images to the fused image.

$Q_s$ analyzes the preservation of salient information from the original images in the fused images using local measures.

$$Q_s(a,b,f) = \frac{1}{|W|} \sum_{w \in W} (\lambda(w) Q_O(a,f|w) + (1 - \lambda(w)) Q_O(b,f|w)). \tag{20}$$

where $W$ is the family of all windows and $|W|$ is the cardinality of $W$. $\lambda$ indicates the relative importance of image $a$ compared to image $b$. $Q_O$ is the overall image quality index. A higher $Q_s$ indicates better overall quality.

SSIM compares the structural similarity between two images by considering local patterns, contrast, and luminance. SSIM is defined as follows:

$$SSIM_{X,F} = \sum_{x,f} \frac{2\mu_x \mu_f + C_1}{\mu_x^2 + \mu_f^2 + C_1} \cdot \frac{2\sigma_x \sigma_f + C_2}{\sigma_x^2 + \sigma_f^2 + C_2} \cdot \frac{\sigma_{xf} + C_3}{\sigma_x \sigma_f + C_3}. \tag{21}$$

where $SSIM_{X,F}$ denotes the structural similarity between source image $X$ and fused image $F$; $x$ and $f$ denote the image patches of source and fused images in a sliding window, respectively; $\sigma_{xf}$ denotes the covariance of source and fused images; $\sigma_x$ and $\sigma_f$ denote the standard deviation; $\mu_x$ and $\mu_f$ denote the mean values of source and fused images, respectively. $C_1, C_2,$ and $C_3$ are the parameters used to make the algorithm stable; when $C_1 = C_2 = C_3 = 0$, the $SSIM$ is reduced to the universal image quality index. Thus, the structural similarities between all source images and the fused image can be written as follows:

$$SSIM = SSIM_{A,F} + SSIM_{B,F} \tag{22}$$

where $SSIM_{A,F}$ and $SSIM_{B,F}$ denote the structural similarities between infrared/visible and fused images. A higher SSIM indicates a greater similarity between the images.

### 4.4. Ablation Study

In this paper, we propose a hybrid attention block (HAB) module as the fundamental computational unit for extracting detail features, and a base feature fusion (BFF) module for integrating cross-modality base features. To evaluate the impact of our proposed HAB and BFF modules on the overall model performance, we conducted ablation experiments and employed quantitative metrics for the analysis. In the ablation study of the HAB module, we substituted it with a simple $1 \times 1$ CNN layer to evaluate the impact of its attention mechanism on the feature extraction. Similarly, to validate the effectiveness of the BFF module, the BFF module was replaced with a standalone LT module. The ablation experiments were conducted using the MSRS dataset.

The results of the ablation experiments are summarized in Table 1. Models without the HAB or BFF modules exhibited a decrease in average performance compared with DDFNet-A, indicating the effectiveness of both the proposed HAB and BFF modules. Notably, the model without the HAB demonstrated a significant drop in performance compared to the model without the BFF. This indicate that the proposed HAB module has a relatively greater impact on the fusion results.

**Table 1.** Quantitative results of ablation experiments on the MSRS dataset. **Bold** indicates the best value.

| | EN | MI | VIFF | $Q^{AB/F}$ | AG | FMI | $Q_s$ | SSIM |
|---|---|---|---|---|---|---|---|---|
| w/o HAB | 6.6047 | 2.7698 | 0.9791 | 0.6623 | 3.6369 | 0.8564 | 0.9288 | 1.4446 |
| w/o BFF | 6.5675 | 2.9265 | 0.9821 | 06729 | 3.6299 | 0.8629 | 0.9301 | **1.4548** |
| DDFNet-A | **6.6169** | **3.0101** | **1.0098** | **0.6804** | **3.6803** | **0.8742** | **0.9313** | 1.4497 |

" w/o HAB" refers to the proposed model without the HAB module, while "w/o BFF" indicates the proposed model without the BFF module.

### 4.5. Comparative Experiments and Analysis

To validate the effectiveness of the proposed method, we compared its fusion performance with twelve state-of-the-art (SOTA) methods: DeFusion [67], DenseFuse [68], FusionGAN [42], ReCoNet [69], SwinFuse [70], SDNet [71], RFN-Nest [72], TarDAL [73], U2Fusion [74], FSFusion [33], MPCFusion [75], and BTSFusion [76]. To ensure a fair comparison, we employed the default parameters provided by the original authors for all twelve methods. These comparative experiments were conducted on three datasets: MSRS, TNO, and VIFB. Comparisons included both qualitative and quantitative evaluations. Qualitative evaluations involved visual inspection of the fused images, with specific targets and details of interest highlighted using red and green boxes.

#### 4.5.1. Results on the MSRS Dataset

Qualitative comparison: Figure 4 shows fused images generated by different algorithms across diverse scenes. The fused images generated by SDNet, RFN-Nest, U2Fusion, FSFustion, and MPCFusion in Figure 4a,b lack detail. The tree branches are blurry and the building textures are unclear. The images generated by FusionGAN are of poor quality, characterized by blurriness, unclear boundaries of targets, the presence of artifacts, and insufficient description of detailed textures. The fused images generated by SwinFuse in Figure 4b,c retain the intensity information of the infrared targets; however, the contrast and brightness are excessively low. This leads to the loss of detailed information and makes it challenging to distinguish background buildings and walls. DeFusion, DenseFuse, ReCoNet, TarDAL, and BTSFusion achieved a commendable balance in all scenes, effectively preserving both significant target information and light information. However, they are not without their shortcomings, as some loss of fine texture details and image noise persists. In contrast, the proposed DDFNet-A achieved the best visual results, with clear visibility of pedestrians, ground markers, and building details. Both the target saliency information and fine texture details are well preserved.

Quantitative comparison: Figure 5 presents a quantitative comparison between the proposed method and the twelve SOTA methods on the 20 pairs of images selected from the MSRS dataset. The average experimental results on the MSRS dataset are presented in Table 2, and the fusion performance of DDFNet-A was compared against twelve SOTA methods. This evaluation employs eight metrics to assess the quality of the fused images generated by various methods. DDFNet-A demonstrated outstanding performance in metrics such as EN, MI, VIFF, $Q^{AB/F}$, FMI, and $Q_s$, while achieving the third-best result in AG and SSIM. The highest EN values indicate that the fused images contain more information content and richer details. The highest values in the VIFF suggest better visual quality of the fused images. The highest values of MI and FMI indicate the model's ability to fuse more source image information into the fused image. The highest values of $Q^{AB/\bar{F}}$,

and $Q_s$ signify the model's superior capability to capture salient features and generate images without noise.

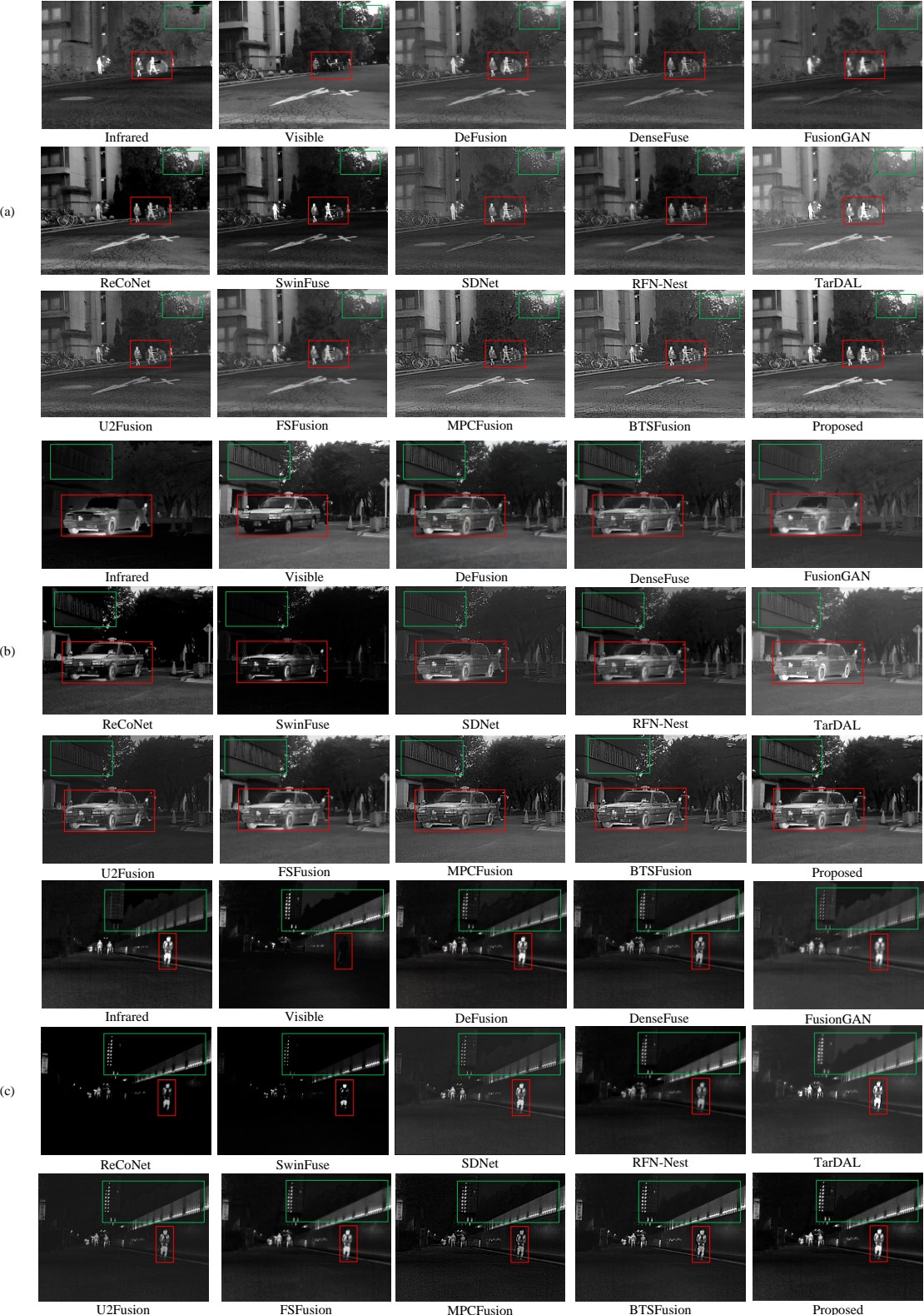

**Figure 4.** Qualitative comparison of selected images from the MSRS dataset: (**a**) 00196D; (**b**) 00131D; and (**c**) 00770N. Some targets and details are annotated with red and green boxes to highlight noteworthy information.

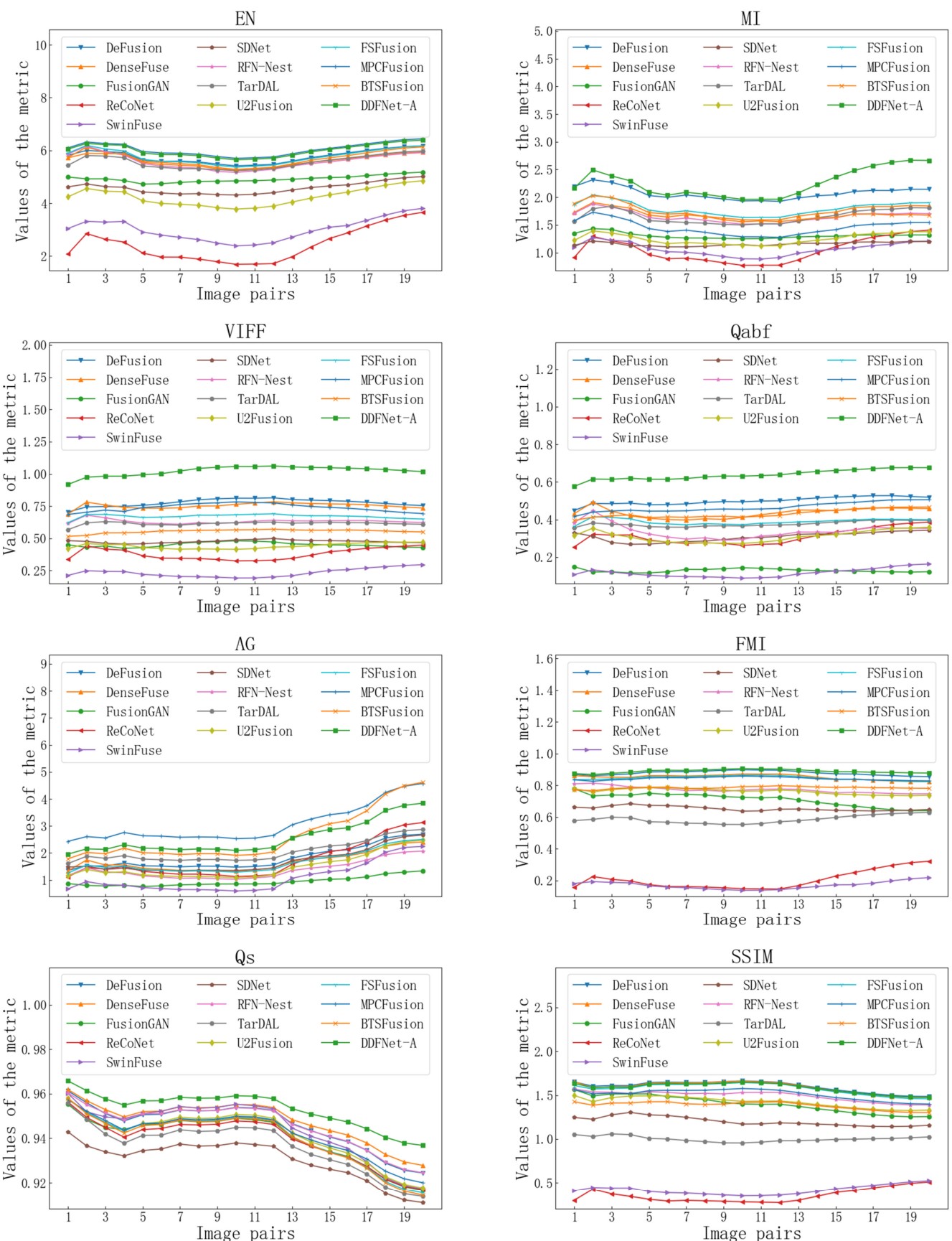

**Figure 5.** Object comparisons of 20 pairs of images selected from the MSRS dataset.

**Table 2.** Quantitative results of comparative experiments on the MSRS dataset. **Bold** indicates the best value, red indicates the second-best value, blue indicates the third-best value.

| | EN | MI | VIFF | $Q^{AB/F}$ | AG | FMI | $Q_s$ | SSIM |
|---|---|---|---|---|---|---|---|---|
| DeFusion | 6.3508 | 2.1585 | 0.7475 | 0.5149 | 2.5959 | 0.8496 | 0.9191 | **1.4709** |
| DenseFuse | 6.1895 | 1.8692 | 0.7713 | 0.4988 | 2.5033 | 0.8230 | 0.9232 | 1.4559 |
| FusionGAN | 5.4314 | 1.3084 | 0.4422 | 0.1401 | 1.4463 | 0.6280 | 0.9098 | 1.2248 |
| ReCoNet | 4.2337 | 1.5821 | 0.4902 | 0.4039 | 2.9897 | 0.3705 | 0.9116 | 0.6120 |
| SwinFuse | 4.2364 | 1.2340 | 0.3599 | 0.1790 | 1.9313 | 0.2649 | 0.9150 | 0.6393 |
| SDNet | 5.2450 | 1.1835 | 0.4984 | 0.3768 | 2.6720 | 0.6858 | 0.9097 | 1.2180 |
| RFN-Nest | 6.1958 | 1.7010 | 0.6558 | 0.3904 | 2.1074 | 0.7591 | 0.9199 | 1.4031 |
| TarDAL | 6.2079 | 1.8307 | 0.6264 | 0.4116 | 2.9350 | 0.6622 | 0.9079 | 1.0784 |
| U2Fusion | 5.2131 | 1.3781 | 0.5161 | 0.3856 | 2.5062 | 0.7537 | 0.9128 | 1.3556 |
| FSFusion | 6.4183 | 1.9874 | 0.6533 | 0.3978 | 2.4405 | 0.8155 | 0.9101 | 1.4411 |
| MPCFusion | 6.6015 | 1.6065 | 0.6849 | 0.5207 | **4.5313** | 0.8230 | 0.9157 | 1.3699 |
| BTSFusion | 6.2913 | 1.5767 | 0.5675 | 0.4899 | 4.4716 | 0.7854 | 0.9096 | 1.2907 |
| DDFNet-A | **6.6169** | **3.0101** | **1.0098** | **0.6804** | 3.6803 | **0.8742** | **0.9313** | 1.4497 |

### 4.5.2. Results on the TNO Dataset

Qualitative comparison: DeFusion, FusionGAN, TarDAL, SDNet, and FSFusion exhibit limitations in preserving the detailed textures in their images. In Figure 6a, the fused images from FusionGAN and SDNet display blurry branch edges. In Figure 6b, the text on the billboard appears unclear in the image generated by TarDAL. In Figure 6c, the building facades in the image generated by DeFusion exhibit a noticeable absence of textural detail. In Figure 6b,c, the images generated by ReCoNet, RFN-Nest, and MPCFusion displays low-intensity, unclear edges of the pedestrian. In Figure 6a,c, SwinFuse effectively balances the intensity information and detailed texture, displaying clear pedestrian features and rich details in branches and buildings. However, in Figure 6b, the image generated by SwinFuse appears darker with a low intensity. Furthermore, the images generated by DenseFuse, U2Fusion, BTSFusion, and DDFNet-A achieve superior fusion results. In particular, DDFNet-A stands out for its exceptional ability to capture fine details without introducing noise.

Quantitative comparison: Figure 7 presents a quantitative comparison between the proposed method and the twelve SOTA methods on the TNO dataset. The average experimental results are listed in Table 3. DDFNet-A demonstrated exceptional performance in metrics such as EN, MI, VIFF,$Q^{AB/F}$, FMI, and $Q_s$, demonstrating minimal information loss during the fusion process, superior visual results, and leading fusion performance.

**Table 3.** Quantitative results of comparative experiments on the TNO dataset. **Bold** indicates the best value, red indicates the second-best value, blue indicates the third-best value.

| | EN | MI | VIFF | $Q^{AB/F}$ | AG | FMI | $Q_s$ | SSIM |
|---|---|---|---|---|---|---|---|---|
| DeFusion | 6.5821 | 1.7573 | 0.5528 | 0.3590 | 2.6747 | 0.7871 | 0.9025 | **1.4698** |
| DenseFuse | 6.7783 | 1.6345 | 0.6309 | 0.4449 | 3.4395 | 0.8045 | 0.9071 | 1.4574 |
| FusionGAN | 6.4803 | 1.6277 | 0.4182 | 0.2244 | 2.3625 | 0.6539 | 0.8893 | 1.2785 |
| ReCoNet | 6.6775 | 1.7181 | 0.5307 | 0.3728 | 3.3534 | 0.7182 | 0.8944 | 1.3202 |
| SwinFuse | 6.9037 | 1.6749 | 0.6394 | 0.4275 | 4.7493 | 0.6851 | 0.8921 | 1.2621 |
| SDNet | 6.6401 | 1.5157 | 0.5569 | 0.4374 | 4.6379 | 0.7803 | 0.8997 | 1.3794 |
| RFN-Nest | 6.8919 | 1.5043 | 0.5414 | 0.3363 | 2.6452 | 0.7417 | 0.9035 | 1.3849 |
| TarDAL | 6.7409 | 1.9312 | 0.5522 | 0.4051 | 3.9971 | 0.7551 | 0.8775 | 1.3681 |
| U2Fusion | 6.6527 | 1.3600 | 0.5633 | 0.4465 | 4.1954 | 0.7976 | 0.8918 | 1.4169 |
| FSFusion | 6.9534 | 1.8630 | 0.5531 | 0.3256 | 2.8183 | 0.7315 | 0.8978 | 1.3615 |
| MPCFusion | 6.9778 | 1.4407 | 0.5226 | 0.4249 | 6.0147 | 0.7727 | 0.8927 | 1.3103 |
| BTSFusion | 6.8489 | 1.2582 | 0.5008 | 0.4167 | **6.0484** | 0.7707 | 0.8838 | 1.3054 |
| DDFNet-A | **7.1217** | **2.1620** | **0.7739** | **0.5426** | 4.8858 | **0.8129** | **0.9079** | 1.3894 |

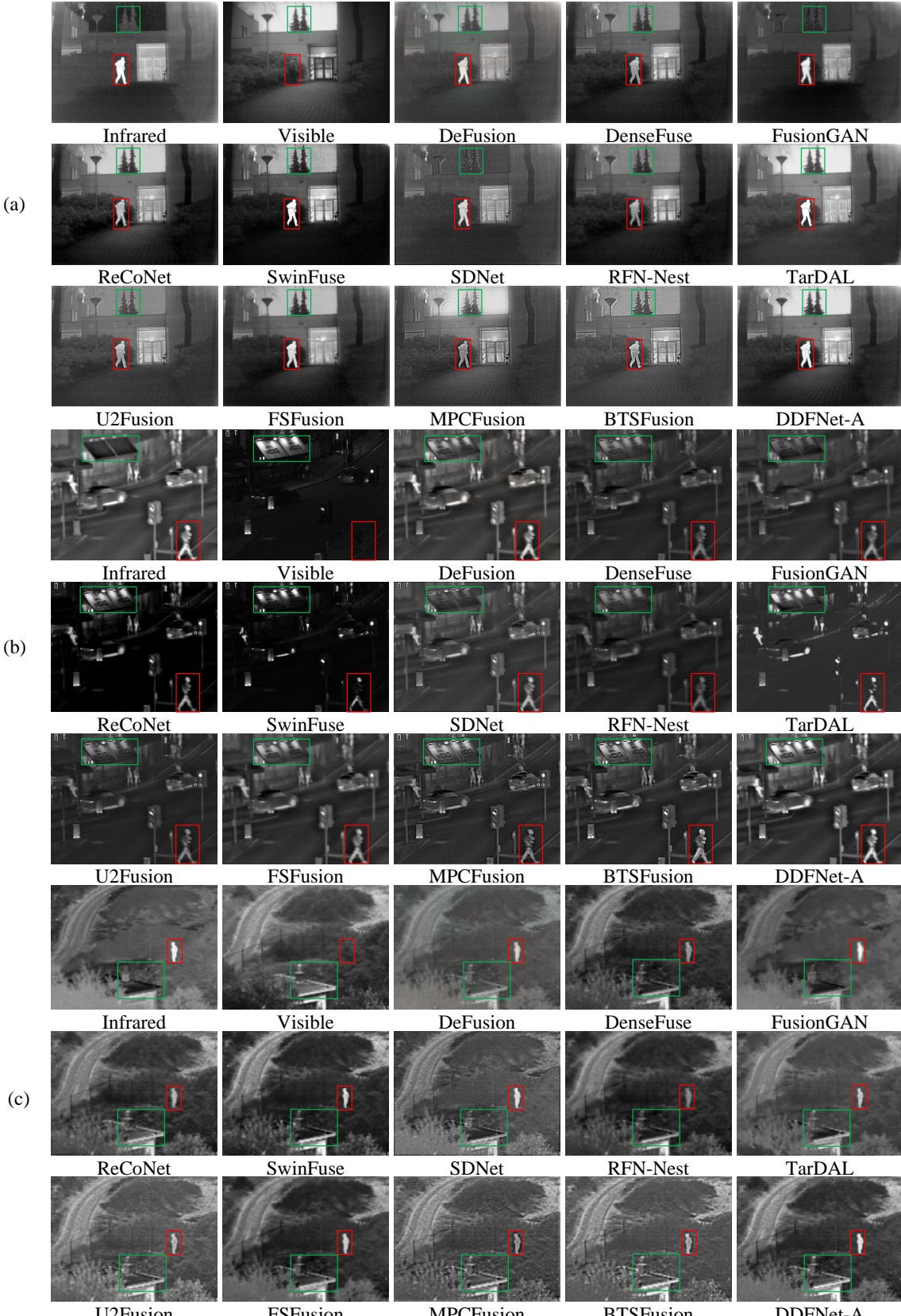

**Figure 6.** Qualitative comparison of selected images from the TNO dataset: (**a**) Kaptein 1123; (**b**) Street; and (**c**) Nato camp. Some targets and details are annotated with red and green boxes to highlight noteworthy information.

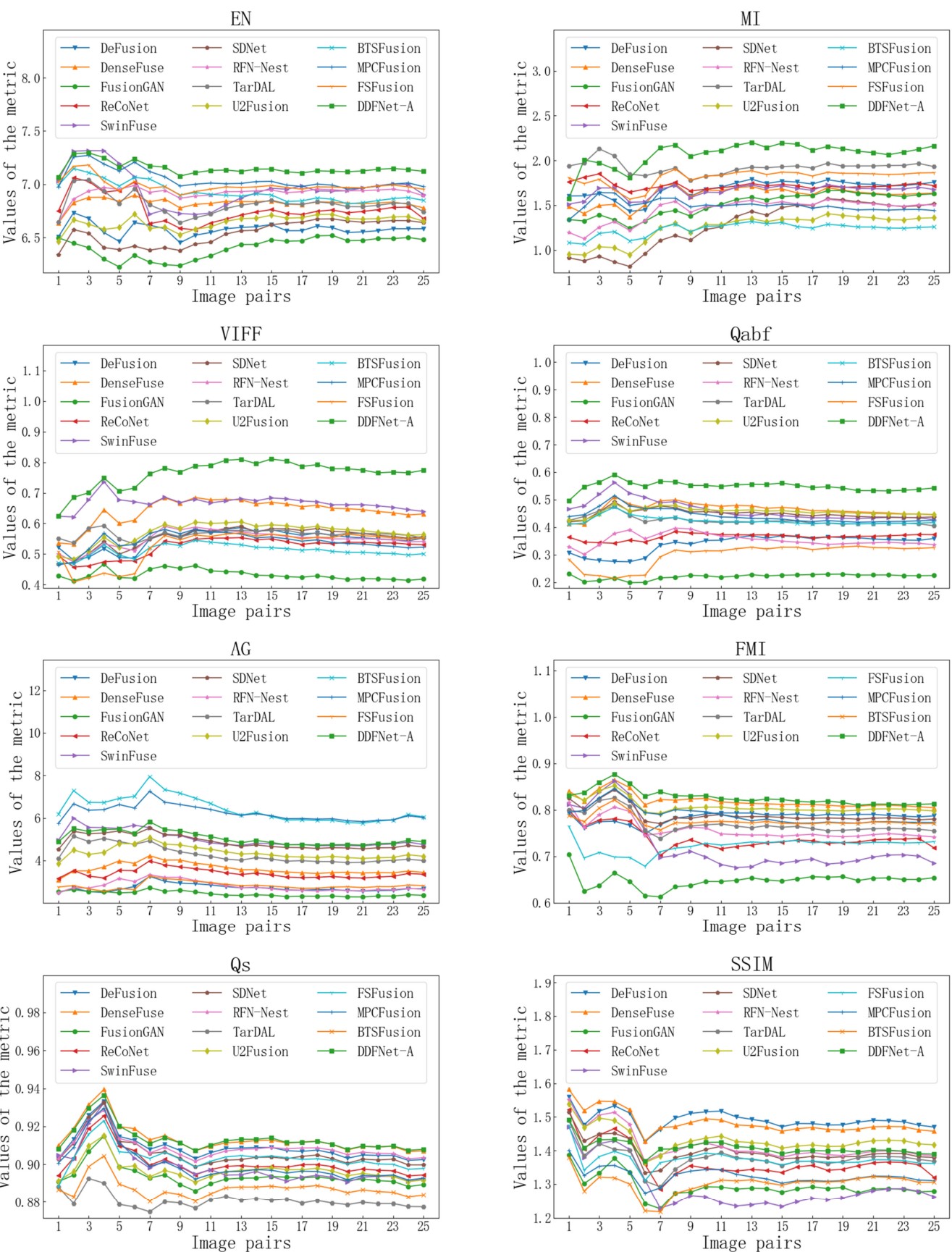

**Figure 7.** Object comparisons of 25 pairs of images selected from the TNO dataset.

### 4.5.3. Results on the VIFB Dataset

Qualitative comparison: The images generated by SDNet and TarDAL lack detail, with blurry buildings and tree branches in Figure 8b,c. FusionGAN and BTSFusion produced fused images with significant artifacts, making it difficult to distinguish details of pedestrians and backgrounds in Figure 8. In the images from DenseFuse, RFN-Nest, and U2Fusion, target intensity information is lost. In Figure 8a, pedestrians blend with shadows in the image from RFN-Nest. In Figure 8b,c, pedestrian features are not prominent enough, and details are blurred in the images from DenseFuse, U2Fusion, FSFusion and MPCFusion. The images generated by SwinFuse lack contrast. In Figure 8a, it is difficult to distinguish pedestrians from shadows, resulting in poor visual effects. In Figure 8c, the details of the branches are lost, disrupted by noise from the sky. DenseFuse, ReCoNet, and DDFNet-A demonstrate strong fusion capabilities. Their fused images retain key information from the source images, such as pedestrians, vehicles, and buildings. Among them, DDFNet-A preserves both significant target information and scene details while presenting superior visual effects.

Quantitative comparison: Figure 9 presents a quantitative comparison on the VIFB dataset. The average experimental results are presented in Table 4. DDFNet-A outperformed twelve SOTA methods on the VIFB dataset across eight metrics, showing exceptional performance in MI, VIFF, $Q^{AB/F}$, FMI, and $Q_s$, and ranking third in EN, AG, and SSIM. High scores in all of these metrics indicate that DDFNet-A effectively combines information from source images while preserving visual quality, removing noise, and capturing important features.

**Table 4.** Quantitative results of comparative experiments on the VIFB dataset. **Bold** indicates the best value, red indicates the second-best value, blue indicates the third-best value.

|  | EN | MI | VIFF | $Q^{AB/F}$ | AG | FMI | $Q_s$ | SSIM |
|---|---|---|---|---|---|---|---|---|
| DeFusion | 6.5712 | 2.0439 | 0.5787 | 0.3682 | 3.1811 | 0.7697 | 0.8996 | **1.4831** |
| DenseFuse | 6.9296 | 2.0730 | 0.6406 | 0.4794 | 4.0765 | 0.7994 | 0.8991 | 1.4572 |
| FusionGAN | 6.2852 | 1.6805 | 0.4352 | 0.2348 | 2.7183 | 0.6738 | 0.8911 | 1.3499 |
| ReCoNet | **7.0985** | 2.2150 | 0.6187 | 0.4861 | 4.5147 | 0.7842 | 0.8933 | 1.3841 |
| SwinFuse | 7.0740 | 2.4317 | 0.7428 | 0.5702 | 5.4986 | 0.7829 | 0.8988 | 1.3411 |
| SDNet | 6.5485 | 1.6131 | 0.5380 | 0.5132 | 5.2860 | 0.8019 | 0.8917 | 1.4379 |
| RFN-Nest | 7.0950 | 2.0691 | 0.5941 | 0.4030 | 3.4578 | 0.7766 | 0.8969 | 1.4195 |
| TarDAL | 6.8719 | 2.3674 | 0.5782 | 0.4302 | 4.0870 | 0.7762 | 0.8788 | 1.4267 |
| U2Fusion | 6.8950 | 1.8828 | 0.6270 | 0.5422 | 4.9646 | 0.8195 | 0.8906 | 1.4350 |
| FSFusion | 6.8351 | 2.1459 | 0.5816 | 0.3803 | 3.3717 | 0.7604 | 0.8967 | 1.4549 |
| MPCFusion | 7.0837 | 1.7712 | 0.5859 | 0.5553 | 6.8123 | 0.7979 | 0.8932 | 1.3536 |
| BTSFusion | 6.9426 | 1.7026 | 0.5133 | 0.5042 | **7.1100** | 0.7818 | 0.8775 | 1.3430 |
| DDFNet-A | 7.0925 | **2.4334** | **0.7701** | **0.6423** | 5.6351 | **0.8376** | **0.9095** | 1.4538 |

### 4.6. Computational Efficiency Analysis

To compare computational efficiency, we conducted ten runs of all image fusion methods on the TNO dataset and calculated the average inference time. It is important to note that the experimental setups for these methods vary. DenseFuse, FusionGAN, SDNet, and U2Fusion methods utilize TensorFlow (GPU version), while DeFusion, ReCoNet, SwinFuse, RFN-Nest, TarDAL, FSFusion, MPCFusion, BTSFusion, and DDFNet-A methods use PyTorch (GPU version). Parameters for all compared algorithms are set to their default values as provided by their respective authors. Table 5 presents the average inference time across the ten runs, with BTSFusion achieving the fastest average inference time at 5.8000 s. Our method ranks eighth among the algorithms, trailing BTSFusion, SDNet, RFN-Nest, FSFusion, SwinFuse, DenseFuse, and TarDAL. Despite our algorithm's eighth-place ranking in terms of inference time, our fusion performance remains state-of-the-art.

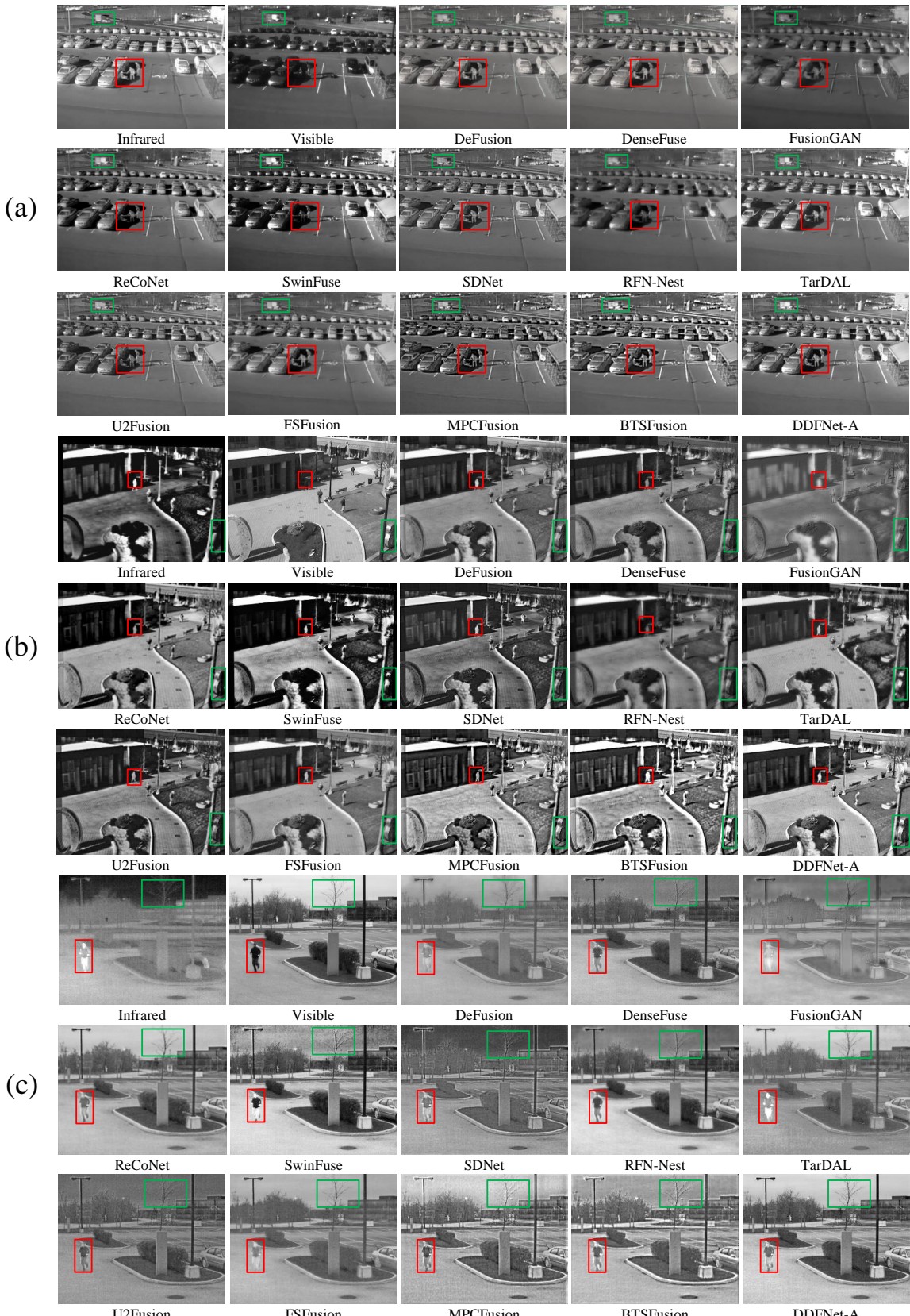

**Figure 8.** Qualitative comparison of selected images from the VIFB dataset: (**a**) fight; (**b**) people shallow; and (**c**) running. Some targets and details are annotated with red and green boxes to highlight noteworthy information.

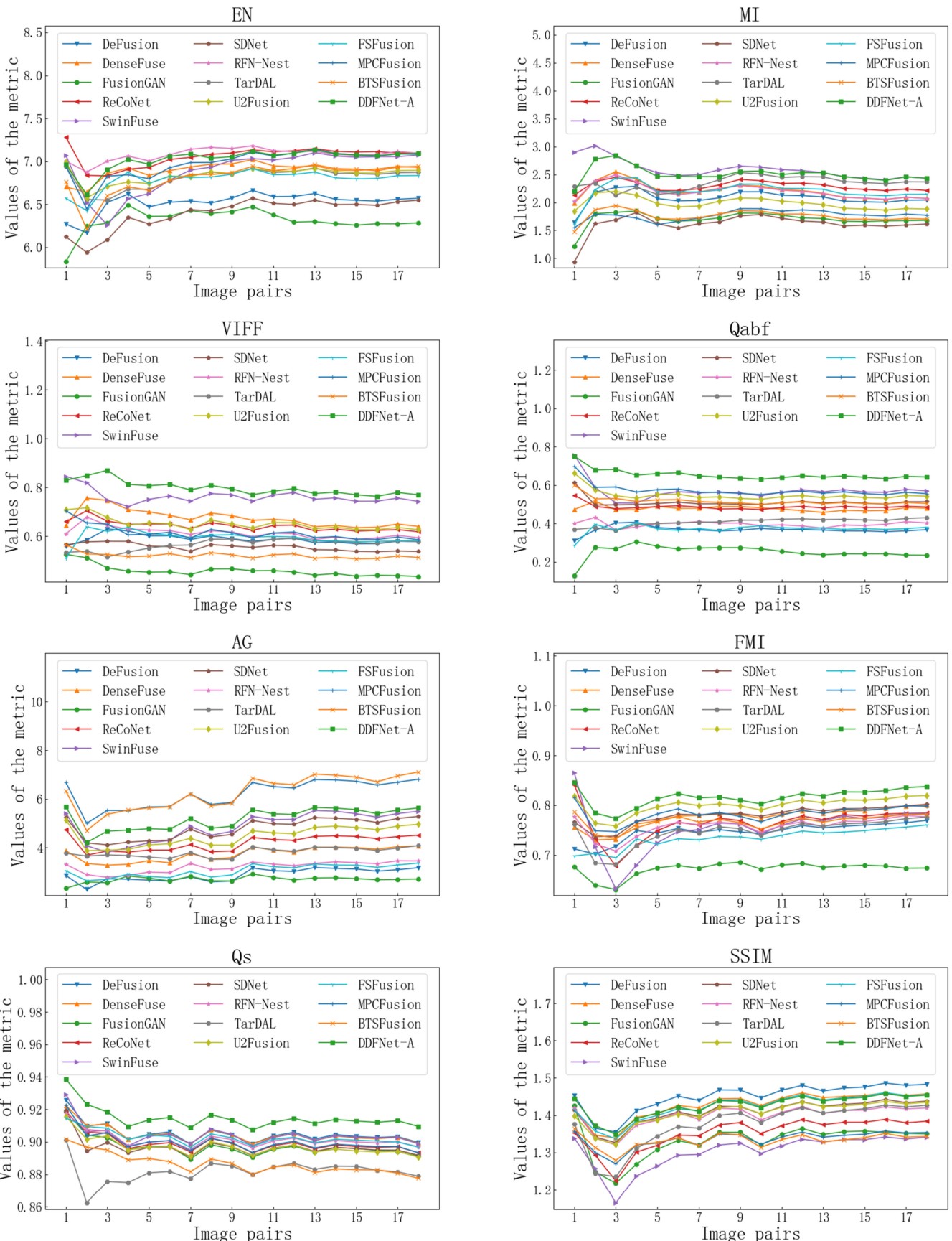

**Figure 9.** Object comparisons of 18 pairs of images selected from the VIFB dataset.

**Table 5.** The average inference time across various methods on the TNO dataset. **Bold** indicates the best value.

| Method (unit: seconds) | DeFusion | DenseFuse | FusionGAN | ReCoNet | SwinFuse | SDNet |
|---|---|---|---|---|---|---|
| | 15.5950 | 14.2975 | 16.9250 | 68.5425 | 9.8625 | 7.2975 |
| RFN-Nest | TarDAL | U2Fusion | FSFusion | MPCFusion | BTSFusion | DDFNet-A |
| 7.5425 | 14.7625 | 18.7800 | 9.4050 | 16.3950 | **5.8000** | 14.9650 |

## 5. Discussion

In general, DDFNet-A outperforms 12 comparison methods in terms of fusion performance on three datasets, demonstrating the effectiveness and robustness of the proposed method. While the proposed method does not significantly outperform other methods in some individual metrics, it is able to balance the quality of fused images across multiple aspects, leading to superior performance in most metrics. This highlights the advancement of the proposed method in various aspects such as information content, preservation of source image information, and visual quality, showcasing its versatility.

## 6. Generalization Experiments on Multi-Focus Image Fusion

Multi-focus image fusion can generate clear images of both distant and near scenes, and there are already some solutions available [67,70,74,77–79]. In order to verify the generalization of the proposed method, extensive experiments were conducted on multi-focus images in this section. The Lytro dataset of color multi-focus images and the Grayscale dataset of gray multi-focus images were selected. Figure 10 shows the remote sensing fusion images obtained using this method. Figure 10a–c, respectively, represent far-focus images, near-focus images, and fusion images.

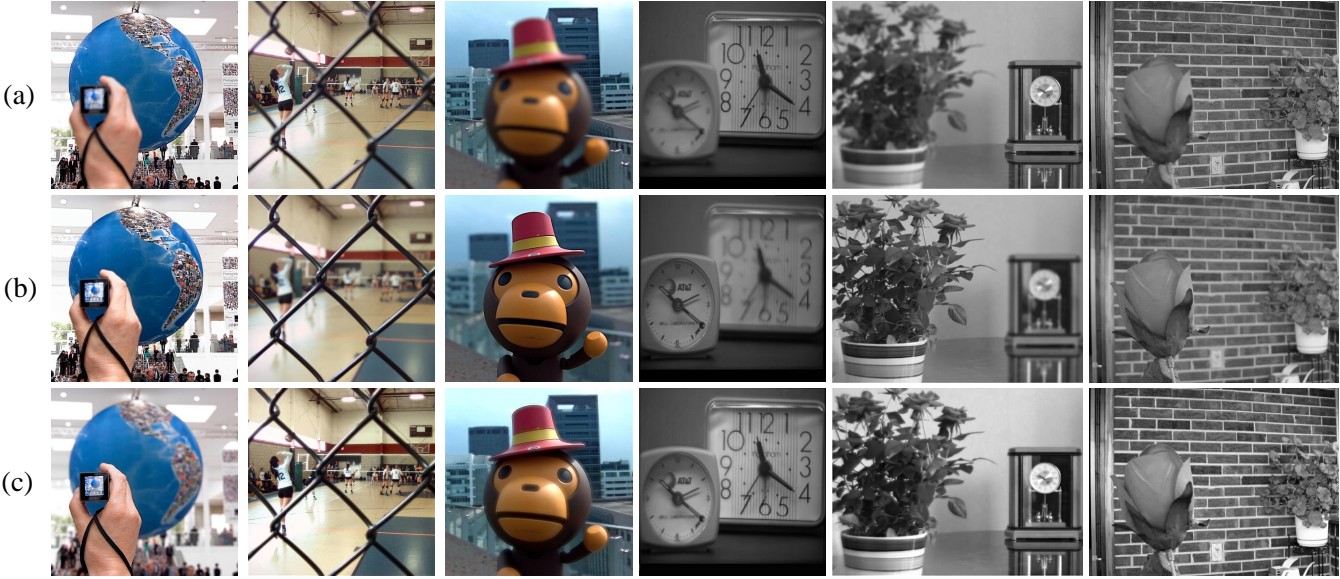

**Figure 10.** Qualitative comparison of selected images from the Lytro and Grayscale dataset: (**a**) far-focus images; (**b**) near-focus images; and (**c**) fused images.

The fusion results show that the fusion image simultaneously retains the image details of distant- and near-focus scenes. This significantly improves the richness of information and the clarity of the fusion image, verifying the robust generalization ability of the proposed method for multi-focus image fusion tasks.

## 7. Conclusions

This study proposes DDFNet-A, a novel attention-based dual-branch feature decomposition fusion network for infrared and visible image fusion. DDFNet-A considers the inherent modality characteristics of infrared and visible images by decomposing them into modality-commonality (base) features and modality-distinctiveness (detail) features. DDFNet-A then performs different fusion strategies on these base and detail features extracted from infrared and visible images. The fused base and detail features were concatenated to generate the fused images. The proposed hybrid attention block (HAB) enhances the extraction ability of detail features across various dimensions, including the channel, frequency, and spatial aspects. Additionally, the proposed base feature fusion (BFF) module utilizes a multi-stage strategy to integrates the base features extracted from the infrared and visible images. The results of the ablation experiment results validated the effectiveness of these modules.

The effectiveness of DDFNet-A was validated on three infrared and visible image datasets, demonstrating its strong performance in complex IVIF scenes. Compared to twelve state-of-the-art methods, DDFNet-A achieved superior fusion quality and efficiency across eight metrics, showcasing its advantages.

Our experiments on multi-focus images confirm that DDFNet-A can be effectively applied to multi-focus fusion scenarios. In our future work, we will extend the evaluation of DDFNet-A's performance across various potential application scenarios, including medical image fusion and multi-exposure image fusion. Furthermore, we will explore the utilization of fused image techniques to enhance the efficacy of other visual tasks, such as object detection and image segmentation.

**Author Contributions:** Conceptualization, Q.W.; methodology, Q.W.; validation, Q.W. and X.J.; formal analysis, Q.W.; investigation, Q.S. and M.Y.; data curation, Q.W. and B.Z.; writing—original draft preparation, Q.W.; writing—review and editing, Y.L.; supervision, Y.L. All authors have read and agreed to the published version of the manuscript.

**Funding:** This work was supported in part by the National Natural Science Foundation of China under Grant #62176247 and research grant #2020-JCJQ-ZD-057-00. It was also supported by the Fundamental Research Funds for the Central Universities.

**Data Availability Statement:** The MSRS dataset is available at https://github.com/Linfeng-Tang/MSRS (accessed on 19 March 2024). The TNO dataset can be download from https://figshare.com/articles/dataset/TNO_Image_Fusion_Dataset/1008029 (accessed on 19 March 2024). The VIFB dataset can be download from https://github.com/xingchenzhang/VIFB/tree/master (accessed on 19 March 2024), The Lytro and Grayscale dataset can be download from https://codeload.github.com/yuliu316316/MFIF/zip/refs/heads/master (accessed on 9 May 2024).

**Conflicts of Interest:** The authors declare no conflict of interest.

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
