# Peer review of "DDFNet-A: Attention-Based Dual-Branch Feature Decomposition Fusion Network for Infrared and Visible Image Fusion"

_remotesensing, doi:10.3390/rs16101795_

Round 1
Reviewer 1 Report
Comments and Suggestions for Authors
The manuscript entitled “DDFNet-A: Attention-Based Dual-Branch Feature Decomposition Fusion Network for Infrared and Visible Image Fusion” presents a new approach for infrared and visible image fusion.
The paper is written well. However, the following comments would improve the quality of the paper.
Fig 2, seems that the sigmoid and multiplication symbols should be replayed with each other.
Implementation details of the proposed method are missing.
Quantitative results of comparing different methods showed the difference between the evaluation metrics are very close and cannot prove the superior quality of the proposed method. This should be discussed in detail.
Author Response
Thanks very much for your valuable comments on our paper, The manuscript has certainly benefited from these insightful revision suggestions. In this revision, we have revised the Paper: remotesensing-2952007 "DDFNet-A: Attention-Based Dual-Branch Feature Decomposition Fusion Network for Infrared and Visible Image Fusion" according to the reviewers’ comments and made a point-by-point response to these comments. Below are the changes we have made in our revised paper as well as responses to your comments.

Reviewer 2 Report
Comments and Suggestions for Authors
This manuscript proposed the attention-based dual-branch feature decomposition fusion network (DDFNet-A) to reconstruct the fused images. The proposed method can improve high-frequency feature extraction ability and enhance the low-frequency feature fusion ability. The description of method is clear. However, the abstract of the paper lacks some important information. For example, what indicator is used for evaluation and what’s the accuracy of the fusion result? There are also some other issues. Hence, I cannot recommend this paper for publication.
The detailed comments are as below:
1) The full name of the acronym “DDFNet-A” haven’t been given.
2) What’s the accuracy of the fusion method? there are no descriptions in the abstract.
3) The introduction should present the applications of the fusion of infrared and visible images, which can make the proposed method meaningful.
4) in Section 4.2, It’s better to present the formulas of these evaluation indicators, and introduce the possible range of the indicators.
5) In my opinion, the introduction of implementing the experiments and the evaluation indicators can be moved to the method section.
Author Response

(The authors gave the same response as above.)

Reviewer 3 Report
Comments and Suggestions for Authors
The paper presents a
- Please check the correctness of Eq. 6, and also try to simplify its representation.
- Add a brief introduction of the datasets. A subsection in Section 4 is recommended.
- The readers would appreciate adding an algorithm of the proposed method.
- Please improve the quality of the graphs in Figs. 6 and 8.
- The results achieved by the presented method are better than SOTA. The authors can consider using more MIF algorithms for comparisons.
- Provide a detailed computational complexity analysis of the proposed method. A comparison with SOTA would be appreciated.
- Can this approach be used for color image fusion? Please discuss and if possible present a few results using Lytro Dataset and Grayscale Dataset.
There are only 2 references from the year 2023. Please update the literature review adding recent advancements on the topic.
Author Response

(The authors gave the same response as above.)

Reviewer 4 Report
Comments and Suggestions for Authors
The authors propose DDFNet-A, an attention-based dual-branch method that decomposes both infrared and visible images into low and high frequency features. The low frequency features capture commonalities across modalities, while the high frequency features highlight the distinctiveness of each modality. This proposed method is interesting, and the experiments are well-conducted on different datasets, demonstrating its superiority over other methods.
However, the paper lacks references to very recent papers (from 2023 and 2024) in both the related works section and the experiments section. Could you please consider adding more recent comparative methods in both sections?
Author Response

(The authors gave the same response as above.)

Round 2
Reviewer 1 Report
Comments and Suggestions for Authors
The paper is revised based on the given comments.
Reviewer 2 Report
Comments and Suggestions for Authors
I think this manuscript is qualified to be published on Remote Sensing now.
Reviewer 3 Report
Comments and Suggestions for Authors
The reviewer appreciates the author for improving the quality of the paper. All his comments are adequately answered in the revised manuscript, and he has no more comments. The manuscript can be accepted.
Reviewer 4 Report
Comments and Suggestions for Authors
The authors have taken the suggestion on board. Thank you.